# Accurate Prediction of Epigenetic Multi-Targets with Graph Neural Network-Based Feature Extraction

**DOI:** 10.3390/ijms232113347

**Published:** 2022-11-01

**Authors:** Yishu Wang, Juan Qi, Xiaomin Chen

**Affiliations:** School of Mathematics and Statistics, University of Science and Technology Beijing, Beijing 100083, China

**Keywords:** graph neural network, XGBoost, DNA methylation, histone modification, multi-target prediction, drug discovery

## Abstract

Epigenetics, referring to genetic modifications that change gene expression, but which are not encoded in DNA, has been shown to be related to oncology, with the potential to influence associated treatments. As such, epigenetic drugs comprise an important new field in cancer therapy; however, drug development is a high-cost and time-consuming procedure. Different epigenetic modifications, such as mutations in DNA methyltransferase and somatic mutations in core histone genes that lead to a global loss of the histone modifications, have innumerable relationships. In this article, we propose a graph neural network-based model for the extraction of molecular features, thus reducing the computational requirements. Through integration with a popular and efficient supervised learner, our model achieves higher prediction accuracy in both single- and multi-target tasks and can determine the pleiotropy associated with drugs, providing theoretical support for drug combination and discovery research.

## 1. Introduction

Over time, the epigenetic modification (including DNA methylation, histone modification, and microRNAs) of genes has played an increasingly important role in the study of tumor genesis and development, especially DNA methylation and histone modifications. DNA methylation is regulated by DNMT and, as one of the most important genetic modification modes, regulates gene transcription levels related to tumor occurrence by inactivating tumor suppressor genes through promoter methylation [1]. Histone modification mainly consists of histone acetylation regulated by histone acetylation transferases (HATs) and deacetylation enzymes (HDACs), besides histone methylation regulated by histone methyltransferases (HMTs) and histone demethylase (HDMs). Studies have shown that histone deacetylation and methylation usually play a role in gene transcription inhibition together with DNA methylation; almost all types of human tumors have abnormal histone modification and DNA methylation. Notably, many tumor suppressor genes are silenced by DNA methylation or histone demethylation in cancers [2,3].

Recently, due to the development of epigenetic research, epigenetic drugs have become a new field, which, differing from the traditional drug mechanisms, are developed from the gene regulation level. To date, three batches of drugs utilizing epigenetic targets have been released, including DNMT inhibitors, HDAC inhibitors, and HMT (targeting EZH2) inhibitors. However, drug development is a costly and time-consuming procedure. Furthermore, the diverse range of molecular mechanisms used by cancer cells to alter epigenetic patterns poses various issues. Different epigenetic modifications, such as mutations in metabolic enzymes, which regulate histone and DNA demethylation, and somatic mutations in core histone genes, which lead to a global loss of histone modifications, have innumerable links [4]. Furthermore, more and more researchers have shown that DNA methylation and histone modification have a certain degree of correlation. For instance, a specific transcription inhibitor MeCP2 has been found to co-exist in one complex with the histone deacetylase (HDAC) in cells [5], and DNMT1 has proven capable of binding to HDAC by identifying the specific binding site. Moreover, the relationship between DNA methylation and histone methylation also has been found gradually [6,7,8,9,10,11,12,13,14].

Therefore, due to the complex relationship between DNA methylation and histone modifications, a single epigenetic target in drug discovery is not significant enough for cancer research. On the contrary, the design of inhibitors should consider multiple targets. Recent research has demonstrated that one of the dual inhibitors of LSD1 (lysine-specific histone demethylase) and HDACs, corin, provided more effective and sustained inhibition of the REST co-repressor complex than existing HDAC inhibitors, and more potently inhibited melanoma proliferation [15]. However, the identification of molecules that act on multiple targets simultaneously through experimental evaluation is costly and time-consuming. The ongoing development of molecule libraries that bind to a specific reader, writer, and eraser domains, such as ChEMBL [16] and the Therapeutic Target Database [17], has opened up the opportunity to construct ligand-based models to assist in small-molecule target prediction; some currently available methods include Similarity Ensemble Approach (SEA) [18], Polypharmacology Browser (PPB) [19], and SwissTargetPrediction [20]. However, these methods usually assign the targets for a given small molecule according to the known targets of the most similar ligands in their data sets. As such, unless the similarity of a known ligand is high enough, these methods are less likely to predict the potential targets of small molecules. This points out a need to develop more effective prediction models focused on multi-target fishing to assist researchers in the medicinal chemistry field.

Conventional methods to extract molecular features usually calculate the three-dimensional structures of molecules, then predict their properties using quantitative structure–activity relationship (QSAR) models [21]. Machine learning algorithms, such as support vector machine (SVM) [22], random forest (RF) [23], and artificial neural networks (ANNs) [24], have been used to predict molecular targets. As machine learning models usually adopt traditional molecular fingerprints, such as Extended Connectivity Fingerprints [25], Morgan fingerprints, and so on, large-scale epigenetic target prediction has been limited by their less-effective performance in predicting molecular properties compared to more recently developed deep learning algorithms [26,27]. Moreover, few studies have focused on multi-target prediction to date.

The symmetry of atomic systems suggested that neural networks that can be applied to network graphs can also be applied to molecular models. For this study, we aim to develop an accurate deep learning model for multiple epigenetic target fishing, integrating GNN-based feature extraction and the conventional classifier extreme gradient boosting algorithm (XGBoost) for classification, which is an efficient implementation of the gradient boosting strategy [28,29]. Graph neural networks (GNNs) can achieve major breakthroughs in learning inter-atom connections, including various GNN sub-types, such as graph convolutional networks (GCNs) [30], gated graph neural networks (GGNNs) [31], and directed message-passing neural networks (DMPNNs) [32]. These various GNN sub-types extract molecular features by capturing the inter-mode relationships through message passing between graph nodes.

We compared the performance of target prediction for three integrated predictive models: GCN+XGBoost, GGNN+XGBoost, and DMPNN+XGBoost. The three integrated models intended for constructing binary classification models were applied to predict the compound-protein associations for 10 key targets related to HDAC, 6 targets related to HMT, 1 target related to DNMT, 2 targets related to HAT, and 5 targets related to HDM. Our investigation demonstrates that multi-target fishing for epigenetic modifications is useful in predicting chemical–protein interactions for the discovery of multi-target inhibitors, providing a useful strategy for the discovery of inhibitor drugs aimed at diseases caused by DNA methylation and/or histone modification.

## 2. Results

The workflow is depicted in Figure 1. First, we developed three integrated graph neural network-based models for identifying the target-associated compounds for the five kinds of epigenetic targets. Second, five-fold cross-validation and test set validations were conducted to evaluate the performance of all models. Third, by combining the 24 single-target binary classifiers, we obtained one multi-target classifier. Finally, the MPNN+XGBoost model was employed to predict drug multi-targets for multiple epigenetic targets including DNMT1, HDACs, HMT, HDM, and HAT. Finally, 33 compounds, including 18 approved DNA methylation and histone modification drugs, as well as 15 other compounds, were predicted to be DNA/histone multi-target inhibitors. These results are detailed below:

### 2.1. Experimental Dataset Analysis

As shown in Figure 2, a total of 10 targets related to HDAC, 6 targets related to HDM, 5 targets related to HMT, 1 target related to DNMT, and 2 targets related to HAT were obtained from the Therapeutic Target Database (TTD) [17] and ChEMBL Database [16] (version 23, https://www.ebi.ac.uk/chembl/ (accessed on 1 May 2022)). A total of 24 targets related to DNA methylation and histone modifications were included and distributed. The compiled chemogenomic data set contained 26,318 unique compounds and 28,845 compound–protein associations, with 17,653 of them being labeled as active (as shown in Figure 3), for an average proportion of active compounds of 61.2%. Moreover, there were 18,679 compounds (71%) in the data set that had known associations to a single target, 485 compounds had known associations to more than 4 targets, while only 20 compounds had known associations to at least 10 targets.

### 2.2. Performance Evaluation and Comparison of Different Graph Neural Networks in Single-Target Performance

Three different binary classification models were built for each of the 24 target-associated compound data sets. Internal five-fold cross-validation and external test set validation were conducted. The performances of the three-graph neural network-based classifiers in the 24 single-target predictions are detailed in Table 1. Among the 10 targets of HDAC, 90% (i.e., 9 out of 10) obtained higher performance metrics in the DMPNN model. Targets HDAC1, HDAC6, and SIRT3 achieved better BA; in other words, our algorithm predicted almost all of the right compound–protein associations on these targets. Among all five targets of HMT, DMPNN achieved the best performance, whereas PRKCB and EHMT2 were the most sensitive targets of HMT. However, two out of four targets (KDM6B and KDM5A) achieved the best BA and MCC with the GCN model. For the targets of HAT and DNMT, the DMPNN model achieved a better performance. These results suggest the DMPNN+XGBoost algorithm as the best combination to derive binary classifiers for the current sets of DNA and histone modification targets studied. Overall, most of the targets in the three models performed well in the single-target prediction task, obtaining mean MCC and F1 scores higher than 0.5, and mean BA higher than 0.7. 

Although MCC, F1 score, and BA are well-suited metrics for model performance, in a practical medical chemistry application, the correct identification of active compounds is usually more important than the correct identification of inactive ones. Therefore, the performance of the three models was studied in terms of precision and recall, as detailed in Table 2. A lower number of compounds predicted as active was associated with a lower recovery of known active compounds. A high recall (TPR) indicates high precision in the prediction of active compounds. Most targets showed mean values of precision and recall higher than 0.6 and 0.8, respectively, suggesting high reliability with regard to the prediction of active compounds.

Moreover, in order to demonstrate the superior feature extraction ability of the graph neural network, we also compared the combination of DMPNN features and the supervised learner (XGBoost) with that of the popular Morgan fingerprint features and XGBoost. Figure 4 shows that the DMPNN-extracted features achieved better AUC performance on all data sets.

### 2.3. Multi-Target Validation

In order to find the multi-target inhibitors of DNA and histone modifications, 24 classifiers were constructed to predict compounds that were active against DNA and histone modifications. First, we compared the global performance of three models and the Morgan fingerprint, when evaluated on the compound data set in which compounds had at least two known active protein targets. In this case, only the correct identification of active compounds (TPR), the correct identification of inactive compounds, and the false discovery rate (FDR) were calculated, considering only the predictions with a truly known label. The performance results of the three models are summarized in Figure 5. Under this validation strategy, DMPNN likewise achieved better multi-target performance, with TPR higher than 0.7, while the NPV was higher than 0.8. 

Hence, the DMPNN+XGBoost multi-target classification model was employed as our DNA methylation and histone modification multi-target predicting model.

### 2.4. Retrospective Identification of Multi-Targets

To verify the application of our graph neural network-based multi-target classifier, DMPNN+XGBoost was developed to predict and explore the multiple bioactivities of approved drugs and new compounds that target DNA methylation and histone modification-related proteins. We adopted 18 approved drugs that target DNA and histone modifications, including 7 drugs targeting DNMT1, 2 drugs targeting HAT, 4 drugs targeting HMT, 1 drug targeting HDM, and 4 drugs targeting HDAC, as well as 15 new compounds with known activity targeting at least one of the 8 most sensitive targets considered in our study (i.e., HDAC6, HDAC1, PRKCB, EHMT2, KEM6B, KDM5A, CREBBP, and DNMT1). 

Case1: Prediction and analysis of polypharmacology of known drugs.

The detailed polypharmacology prediction results for the known drugs are presented in Table 3 and Figure 6, where the hypergeometric test was employed to determine the statistically significant targets (i.e., HDAC, HAT, HDM, HMT, DNMT) of every drug. The results indicate that all known targets of the drugs were predicted correctly, where the colors that represented objects in this table are consistent and ‘Sig’ is colored red. Moreover, we also determined the pleiotropy of these DNA methylation and histone modification inhibitors, which, on the other hand, allowed for certification of evaluation in the multi-target drugs. In particular, the approved DNA methyltransferase inhibitors azacitidine and decitabine were predicted by our model to significantly target HDAC, which is reasonable, as histone deacetylation has inextricable links with DNA methylation. Harada T. et al. have shown that HDAC3 can regulate DNMT1 expression [33]. Moreover, researchers have demonstrated that a drug combination for inhibition of DNMT and HDAC can block the tumorigenicity of cancer stem-like cells and attenuates mammary Tumor Growth [34]. Inversely, the inhibitors procaine, cephalothin, and procainamide have high target specificity. Furthermore, in our model, the four HDAC inhibitors (belinostat, vorinostat, romidepsin, and panobinostat) were not only predicted with the correct target (HDAC) but also exhibited other active target predictions. For example, the first approved HDAC inhibitor, vorinostat, was also predicted to be active with the DNMT target. Consistently, it has been stated that the JmiC family of lysine demethylases contain an active site iron cation, such that vorinostat was found to inhibit the demethylase JMJD2E [35]. 

Case2: Multi-target prediction of new compounds

A set of 15 compounds (Figure 7) with known activity targeting at least one of the 8 most sensitive targets of our model was retrieved from the ChEMBL database. The prediction results of our multi-target classifier on these 15 compounds are shown in Figure 8, where drugs were likewise clustered based on the predicted active targets. For example, the compound CHEMBL4297494 (name: GSK-3117391), which is under investigation in clinical trials for Rheumatoid Arthritis, presented similar multi-targets as compound CHEMBL8809 (name: raclopride) which is under investigation in a phase 1 clinical trial studying Parkinson’s Disease. Both of them targeted proteins HDAC6, HDAC1, PRKCB, CREBBP, and DNMT. Furthermore, compounds CHEMBL2036482, CHEMBL4297366 (CG-200745), CHEMBL3693786 (citarinostat), and CHEMBL1420319, among others, showed congruous targeting of HDM proteins KDM6B and KDM5A, as well as the HAT protein CREBBP, indicating the correlation between histone demethylation with acetylation.

### 2.5. Molecular Docking Verification

In order to verify the multi-target predicting results of our algorithm, molecular docking analysis was conducted by the software AutoDock Vina [25]. Here we verified the binding sites of the first approved HDAC inhibitor, vorinostat with target protein HDAC1 and the predicted target protein DNMT1 (Figure 9A,B). Then the binding modes of the other two new compounds: citarinostat (which has been approved to be one specific HDAC6 inhibitor), and CHEMBL375661 show the active binding with the protein targets predicted by our algorithm (Figure 9C–F).

## 3. Discussion

The epigenetic modification of genes, including DNA methylation and histone modification, has attracted more and more attention due to its role in the regulation of gene transcription, rather than changing the DNA sequence. In particular, the epigenetic modification of genes has been shown to play an increasingly important role in tumor development; studies have shown that almost all types of human tumors present abnormal histone and DNA methylation, causing the silencing of tumor suppressor genes, and demonstrating that there is a certain degree of correlation between them. Due to developments in chromatin biology, novel drugs directed at chromatin and associated components have been identified, mainly focused on DNA methylation inhibitors and histone deacetylase inhibitors. However, epigenetic changes are complex and anfractuous; for instance, methylation of histone protein H32K9 can (directly or indirectly) influence the DNA methylation mode while, simultaneously, some DNA methylation may follow as a result of histone methylation. Concurrently, among the seven epigenetic anti-cancer drugs, few work well alone in treating cancer; in contrast, combinations of DNA methylation and histone deacetylation inhibitors have yielded more positive results. 

However, traditional drug research, following the idea of trial and error to explore targets, not only has high costs and a long cycle but also may lead to bottlenecks in accuracy; it cannot meet the demand for multi-target prediction. As a branch of machine learning in artificial intelligence, deep learning has been applied in the field of biomedicine, providing enhanced expressive power in identifying, processing, and inferring complex patterns of molecular data. Molecular property prediction is one of the most important problems in drug development; however, most deep learning methods adopt traditional molecular fingerprints, such as Morgan fingerprints, extended connectivity fingerprints, and so on, which do not allow for the extraction of spatial features of molecules. The insufficient extraction of molecular features seriously affected the downstream prediction results. Moreover, epigenetic therapies have broad specific targets of drugs. Therefore, the current trend in epigenetic cancer research is to target two or more proteins, which needs innovative multi-target prediction algorithms.

In this study, we proposed a graph neural network-based molecular feature extraction algorithm, which was constructed by embedding a popular supervised learning algorithm to build multiple classifiers, which can be used to predict epigenetic targets (especially those related to DNA methylation and histone modification). Then, by combining these DNA and histone-related classifiers, a multi-target predictive model was built. The proof-of-principle study demonstrated that the integration of DMPNN and XGBoost may efficiently improve the performance of prediction problems regarding DNA methylation and histone modification targets. Furthermore, the achieved multi-target prediction may serve as foundational guidance for drug combination and development research. The carried performance evaluations demonstrated the directed message-passing neural network extracted more sufficient molecular features than the traditional Morgan fingerprints. Actually, the directed message-passing neural network has been demonstrated can be applied to chemical prediction tasks to learn the characteristics of molecules directly from molecular graphs and are not affected by graph isomorphism.

Although experimental validation of the predictions from all models is required to provide better means of comparison, our findings reinforce the potential usefulness of a tool focused on the prediction of multiple epigenetic targets for drug discovery. Furthermore, molecular docking simulations have been adopted to verify the prediction results of several drug targets on the predicted epigenetic proteins by our algorithms. In the future, we intend to study more molecular docking modes of multi-target compounds of interest, further assisting AI-based drug discovery focused on DNA methylation and histone modification.

## 4. Materials and Methods

### 4.1. Data Set Preparation

The quantitative compound–protein associations related to DNA methylation and histone modifications were extracted from ChEMBL 27 [36] and PubChem [28], in order to build epigenetic target-associated compound data sets, following the criteria: (1) Compounds with an IC_50_, EC_50_, K_i_, or K_d_ lower than or equal to 10 μm were retained as “active”, while those higher than 10 μm were retained as “inactive”; (2) target proteins containing at least 30 compounds were labeled as “active”, while those with at least 30 compounds were labeled as “inactive”. 

### 4.2. Molecular Graph

The simplified molecular-input line-entry specification (SMILES) information of compound molecules, composed of a series of ASCII-encoded strings, cannot be directly input into graph neural network models. Thus, first, we needed to express the SMILES data using a quantized molecular graph. We used the RDKit tool [36] to process these SMILES encoding compounds in order to obtain the associated molecular graphs. 

### 4.3. Multi-Target Fishing Model Generation

In this study, first, we adopted three GNN models to extract the molecular features and then utilized the extreme gradient boosting algorithm (XGBoost) for classification. The states of the graph nodes were updated using the node embedding method, which is described as follows: $h_{i}^{t}=U(h_{i}^{t-1},m_{i}^{t})$, where the *i*th node is updated using the previous state $h_{i}^{t-1}$ and a message state of the interaction term $m_{i}^{t}$ with its neighborhoods. Based on this simplest version of GCN, the GGNN utilizes gate recurrent units (GRUs) in the propagation step, while the directed MPNN (DMPNN) propagates information through directed bonds and generalizes various existing GNNs. By inputting the molecular graph information and the active/inactive information on protein targets, the three GNNs carried out supervised learning, resulting in the feature representation. 

Secondly, the graph representations were loaded as sample features into the supervised learner, XGBoost, to complete the prediction of classifications. XGBoost is an effective implementation of a gradient enhancement strategy. Similar to a decision tree, it uses gradient lifting trees as a weak classifier and integrates strong classifiers by voting or weighted grounding. Moreover, the original loss function, $y_{loss}=\sum_{i}l(y_{i}^{ʹ},y_{i})$, was improved by adding a regular penalty term, as follows: $y_{loss}=\sum_{i}l(y_{i}^{ʹ},y_{i})+\sum_{k}\Omega (f_{k})$, where $f_{k}$ is the weight information of each decision tree in the training process.

Concurrently, in the training process, the Adam optimizer [37] was used, in order to force the loss function as close as possible to the global lowest point in the BP process. This optimizer controlled the step size of the learning rate and the gradient direction according to the first- and second-order momentum and suppressed the stationary phenomenon of the gradient near saddle points to prevent oscillation of the gradient. 

Finally, by combining these 24 single-target binary classifiers, we obtained one multi-target classifier. 

### 4.4. Performance Metrics

To compare the models generated herein in a more global context, cross-validation was adopted for each model, which was used to calculate the performance metrics (Mathews correlation coefficient, MCC; F1; and balanced accuracy, BA), where MCC, F1, and BA are well-suited metrics for model performance estimation in imbalanced data sets, which balance the recall rate with the precision rate.
$$\mathrm{MCC}=\frac{\mathrm{TP}\times \mathrm{TN}-\mathrm{FP}\times \mathrm{FN}}{\sqrt{\left(\mathrm{TP}+\mathrm{FP})(\mathrm{TP}+\mathrm{FN})(\mathrm{TN}+\mathrm{FP})(\mathrm{TN}+\mathrm{FN}\right)}}$$
$$F1=2\times \frac{\mathrm{TP}}{2\mathrm{TP}+\mathrm{FP}+\mathrm{FN}}$$
$$\mathrm{BA}=\frac{0.5\mathrm{TP}}{\mathrm{TP}+\mathrm{FN}}+\frac{0.5\mathrm{TN}}{\mathrm{TN}+\mathrm{FP}}$$
where TP denotes true positives, TN denotes true negatives, FP denotes false positives, and FN denotes false negatives, with positive and negative referring to active and inactive compound labels, respectively.

### 4.5. Experimental Environment

The experiment in this paper was run on an Ubuntu system under the Linux bit 4.15.0-173-generic #182-Ubuntu SMP X86_64 GNU/Linux version, and the running environment was built using Conda. We used Python version 3.8.13 as the programming language, an NVIDIA GeForce RTX 3090 graphics card with CUDA version 11.4, the RDKit 2022.3.1 package for molecular fingerprinting, and Pytorch 1.11.0 version to complete the construction process of the deep learning network.

## 5. Conclusions

In the past, “one protein, one drug, one disease” was the mainstream research model. However, with the development of polypharmacology and multi-target therapy, in addition to huge challenges in processing large amounts of complex data from genomics, proteomics, microarrays, and clinical trials, this has changed. The pharmaceutical sector has modernized using machine learning and deep learning algorithms in drug discovery processes such as peptide synthesis, virtual screening, toxicity prediction, drug monitoring and release, pharmacophore modeling, quantitative structure-activity relationship, multi-pharmacological, physiological activities, and so on. Therefore, intelligent multi-target drug discovery with deep learning algorithms has become an important direction of drug research and development.

Within the field of drug discovery, the first and most important step is to identify appropriate targets (e.g., genes, and proteins) related to the pathophysiology of the disease, then find drugs or drug-like molecules that interfere with those targets. In the era of big data, we have huge chemical databases at our disposal, such as PubChem, ChEMBL, DrugBank, and so on. In this study, in view of epigenetic targets, a significant focus for discovery research, we proposed one multi-target prediction algorithm based on a feature extraction algorithm using the deep learning method. In this algorithm, the graph neural network-based molecular feature extraction algorithm embeds one popular supervised learning algorithm, to build multiple classifiers, predicting the epigenetic (especially, DNA methylation and histone modification) targets. By combing these binary classifiers, one multi-target prediction model was constructed. Our algorithm could extract molecular features sufficiently, thus resulting in efficient prediction results. The prediction results demonstrate that the integrated deep learning model showed high performance in the prediction of epigenetic multi-targets. Furthermore, to illustrate the application of our model, molecular docking was adopted in several small molecules and target proteins, which verified the possibility of our prediction results.

Thus, the results showed that it is possible to design specific inhibitors targeting two or more DNA methylation and histone modification targets simultaneously. In addition, an individual compound was determined to have coinhibitory ability against more than one target. In the future, based on our algorithm results, we will conduct trials of drugs using DNA methylation and histone modification that have been preliminarily screened by our model.

## Figures and Tables

**Figure 1 ijms-23-13347-f001:**
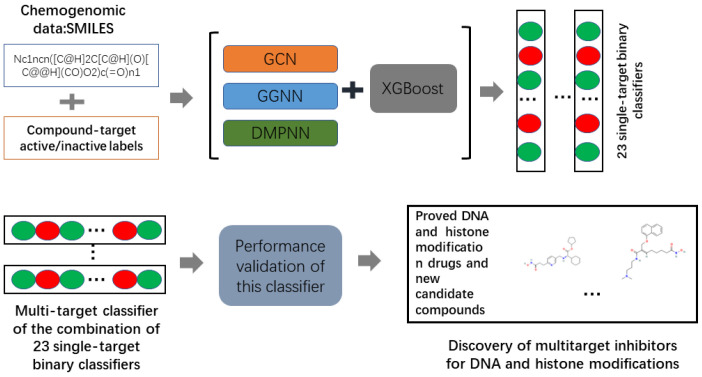
Workflow of our study.

**Figure 2 ijms-23-13347-f002:**
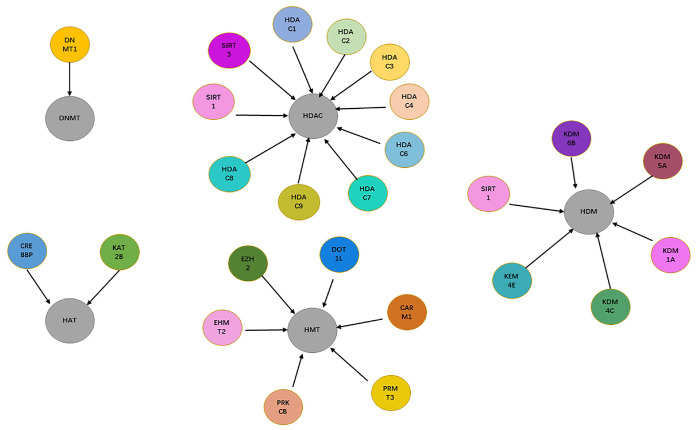
Overview of the target proteins related to epigenetic objects: DNMT, HMT, HAT, HDM, and HDAC.

**Figure 3 ijms-23-13347-f003:**
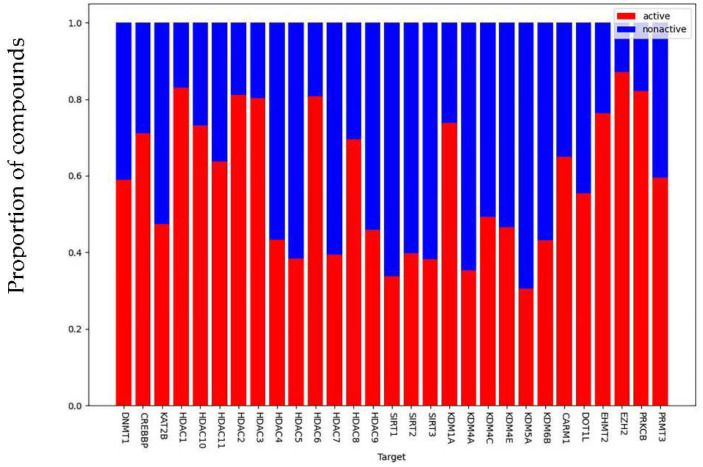
Composition of the target-associated compound data sets. Epigenetic target-associated compound datasets consisted of 693 compounds on average, all 24 compound datasets had different class imbalance levels, showing an average proportion of active compounds of 61.3%, with a maximum of 91.4% of EZH2.

**Figure 4 ijms-23-13347-f004:**
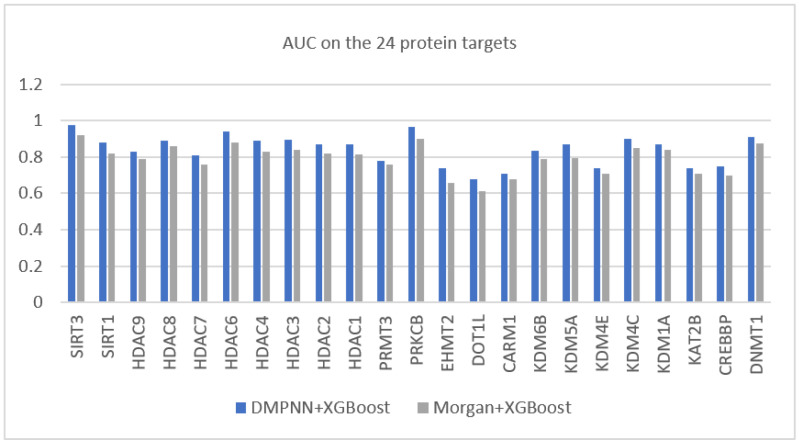
Performance comparison of the XGBoost models using DMPNN features and Morgan fingerprints. The horizontal axis lists the protein targets.

**Figure 5 ijms-23-13347-f005:**
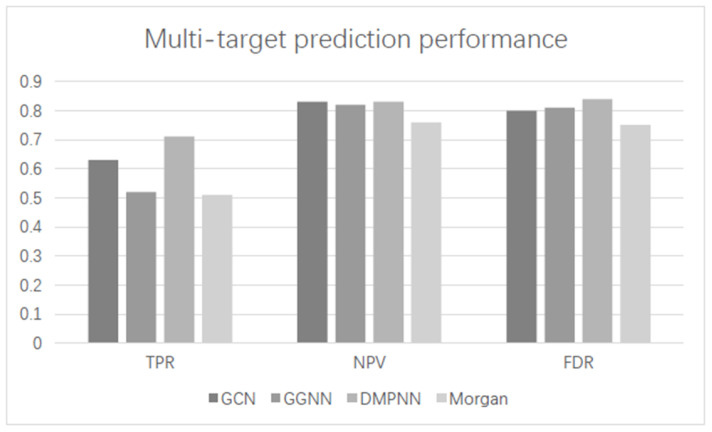
Multi-target Performance comparison with different models. The supervised learning algorithm was XGBoost. TPR, NPV, and FDR for all compounds on the combination classifier.

**Figure 6 ijms-23-13347-f006:**
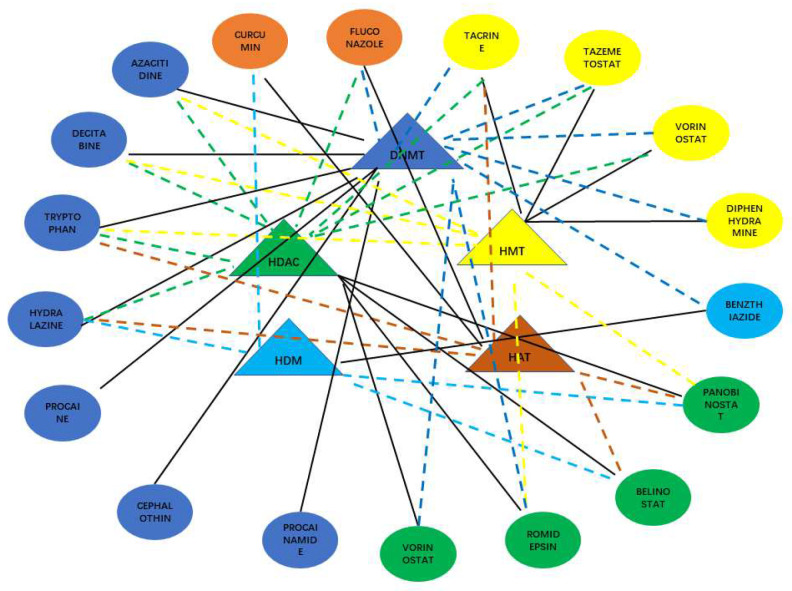
Polypharmacological analysis of 18 known drugs with four epigenetic objects. Different colors represent different epigenetic objects: Blue: DNMT, orange: HAT, yellow: HMT, light blue: HAT, green: HDAC. Sig indicates statistical significance in the Hypergeometric test for all targets in each object. Black lines indicate proven targeting interactions.

**Figure 7 ijms-23-13347-f007:**
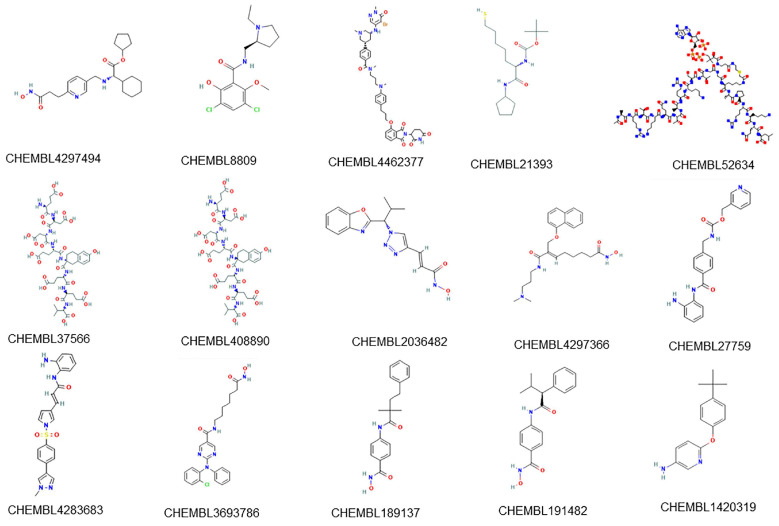
The 2D structures of 15 selected new compounds.

**Figure 8 ijms-23-13347-f008:**
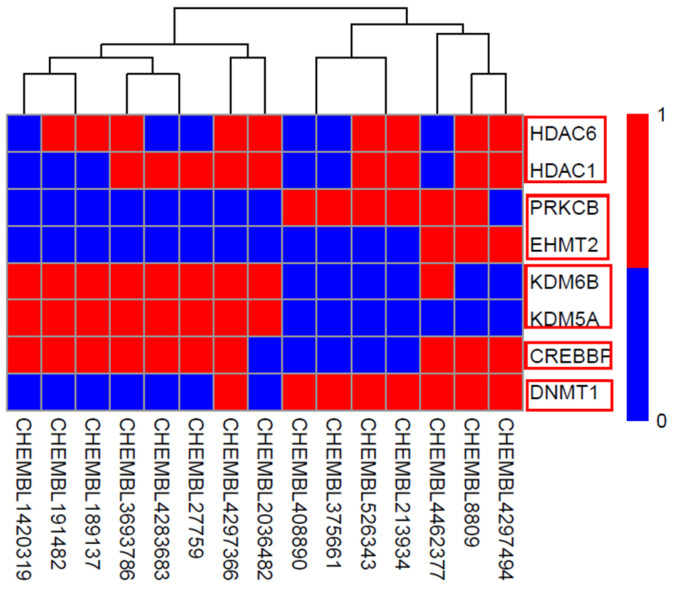
Prediction results of 15 new compounds that were not involved in the generation of our multiple models. The red box represented different objects, such as: DNMT, HDAC, and so on.

**Figure 9 ijms-23-13347-f009:**
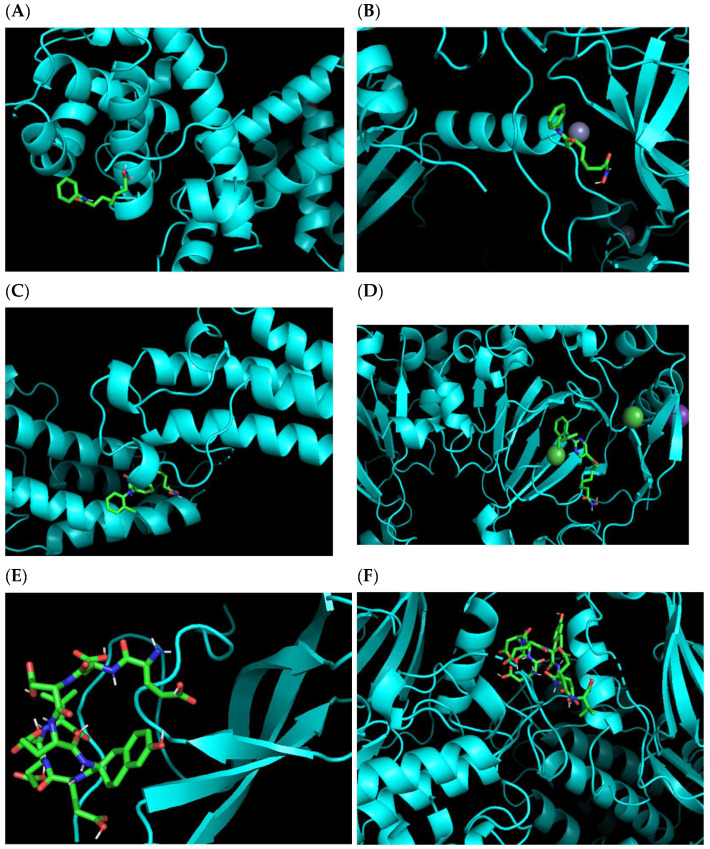
The molecular docking results of the binding mode of three compounds in their respective binding sites. (**A**,**B**) show the binding modes of vorinostat with HDAC1 (PDB: 7SME) and DNMT1 (PDB: 7LMM). (**C**,**D**) show the binding modes of citarinostat with CREBBP (PDB: 7XHE) and KDM6B (PDB: 2XXZ). (**E**,**F**) show the binding modes of compound CHEMBL375661 with DNMT1 (PDB: 7LMM) and PRKCB (PDB: 210E).

**Table 1 ijms-23-13347-t001:** Single-target performance comparison of three GNNs. The supervised learning algorithm was XGBoost. Three performance metrics: MCC, F1, and BA were adopted for these 24 single-target binary classifiers.

Object	Target	GCN			GGNN			DMPNN		
		MCC	F1	BA	MCC	F1	BA	MCC	F1	BA
HDAC	SIRT3	0.516	0.556	1	0.411	0.526	0.833	0.516	0.556	1
	SIRT1	0.667	0.78	0.75	0.624	0.75	0.732	0.602	0.734	0.725
	HDAC9	0.391	0.725	0.735	0.282	0.725	0.644	0.482	0.784	0.75
	HDAC8	0.669	0.896	0.862	0.69	0.901	0.867	0.721	0.913	0.863
	HDAC7	0.283	0.55	0.55	0.33	0.613	0.543	0.436	0.655	0.667
	HDAC6	0.637	0.941	0.93	0.625	0.938	0.93	0.689	0.948	0.94
	HDAC4	0.482	0.714	0.727	0.43	0.64	0.775	0.6	0.767	0.82
	HDAC3	0.55	0.919	0.89	0.56	0.92	0.9	0.6	0.93	0.91
	HDAC2	0.44	0.91	0.86	0.314	0.9	0.84	0.48	0.92	0.88
	HDAC1	0.46	0.92	0.89	0.46	0.922	0.91	0.57	0.94	0.92
HMT	PRMT3	0.315	0.643	0.75	0.167	0.571	0.667	0.667	0.87	0.762
	PRKCB	0.553	0.935	0.906	0.51	0.925	0.901	0.615	0.937	0.934
	EHMT2	0.04	0.61	0.82	0.05	0.88	0.81	0.25	0.9	0.83
	DOT1L	0.27	0.42	0.8	0.285	0.643	0.64	0.247	0.66	0.571
	EZH2	0.403	0.942	0.892	0.323	0.92	0.90	0.32	0.931	0.875
	CARM1	0.12	0.667	0.63	0.34	0.77	0.68	0.13	0.667	0.63
HDM	KDM6B	0.201	0.133	1	0.276	0.522	0.667	0.189	0.6	0.526
	KDM5A	0.341	0.375	0.75	0.434	0.526	0.71	0.167	0.25	0.5
	KDM4E	0.176	0.4	0.6	0.186	0	0	0.28	0.556	0.625
	KDM4C	0.173	0.6	0.437	0.309	0.59	0.59	0.45	0.68	0.64
	KDM1A	0.42	0.89	0.84	0.55	0.9	0.89	0.537	0.891	0.91
HAT	KAT2B	0.19	0.65	0.52	0.24	0.67	0.5	0.267	0.647	0.58
	CREBBP	0.36	0.86	0.77	0.245	0.83	0.77	0.368	0.82	0.83
DNMT	DNMT1	0.316	0.83	0.8	0.438	0.76	0.86	0.55	0.83	0.88

**Table 2 ijms-23-13347-t002:** Single-target performance comparison of three GNNs. The supervised learning algorithm was XGBoost. Results of Precision and Recall for these 24 single-target binary classifiers.

Model	Object	PRECISION	RECALL
GCN	HDAC	0.819 ± 0.091	0.796 ± 0.013
	HMT	0.782 ± 0.11	0.94 ± 0.081
	HDM	0.762 ± 0.19	0.64 ± 0.13
	HAT	0.65 ± 0.15	0.98 ± 0.03
	DNMT	0.88	1
GGNN	HDAC	0.79 ± 0.02	0.766 ± 0.06
	HMT	0.74 ± 0.21	0.91 ± 0.11
	HDM	0.611 ± 0.32	0.58 ± 0.06
	HAT	0.602 ± 0.21	0.975 ± 0.04
	DNMT	0.86	1
DMPNN	HDAC	0.85 ± 0.02	0.95 ± 0.13
	HMT	0.85 ± 0.17	0.945 ± 0.02
	HDM	0.66 ± 0.11	0.71 ± 0.3
	HAT	0.705 ± 0.19	0.99 ± 0.01
	DNMT	0.89	1

**Table 3 ijms-23-13347-t003:** Retrospective prediction of known drugs with their known targets. Different colors represent different epigenetic objects: Blue: DNMT, orange: HAT, yellow: HMT, light blue: HAT, green: HDAC. Sig indicates statistical significance in the Hypergeometric test for all targets in each object (*p*-value < 0.05).

Drug Name	HDAC	HMT	HDM	HAT	DNMT
AZACITIDINE	Sig	No	No	No	Sig
DECITABINE	Sig	No	No	No	Sig
TRYPTOPHAN	No	Sig	Sig	Sig	Sig
HYDRALAZINE	Sig	No	Sig	Sig	Sig
PROCAINE	No	No	No	No	Sig
CEPHALOTHIN	No	No	No	No	Sig
PROCAINAMIDE	No	No	No	No	Sig
CURCUMIN	No	No	Sig	Sig	No
FLUCONAZOLE	Sig	No	No	Sig	Sig
TACRINE	Sig	Sig	No	Sig	Sig
TAZEMETOSTAT	Sig	Sig	No	No	Sig
VORINOSTAT	Sig	Sig	No	No	Sig
DIPHENHYDRAMINE	No	Sig	Sig	Sig	Sig
BENZTHIAZIDE	No	No	Sig	No	Sig
PANOBINOSTAT	Sig	Sig	Sig	Sig	No
BELINOSTAT	Sig	No	Sig	Sig	No
ROMIDEPSIN	Sig	Sig	No	No	Sig
VORINOSTAT	Sig	No	No	No	Sig

## Data Availability

All data in our manuscript and Python codes are available: https://github.com/Yswangustb/DMPNN.

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
