# Peer review of "Accurate Prediction of Epigenetic Multi-Targets with Graph Neural Network-Based Feature Extraction"

_ijms, 2022, doi:10.3390/ijms232113347_

Round 1
Reviewer 1 Report
The authors present a bioinformatics-based study related to potential therapeutic targets of two major epigenetic mechanisms: DNA methylation and histone modifications.
The study is indeed interesting, but a major improvement of the manuscript is needed before it is recommended for publication at IJMS, as follows:
1. The title should be re-written to indicate that this is a bioinformatics-based study. Please use scientific terminology; for example fishing in the title and the text is not scientific.
2. Taking comment 1 into consideration, it is strongly recommended to have this manuscript revised by a native English language speaker.
3. Abbreviations should always be preceded by the full name first (in the introduction DNMT is mentioned without indicating the full name "DNA methyltransferase".
4. MicroRNAs are also considered a major epigenetic mechanism. The authors should mention this mechanism and justify why they chose to assess only DNA methylation and histone acetylation in this study.
5. Since the introduction is extremely extensive, it is strongly recommended to have diagrams to illustrate the DNA methylation and histone modification mechanisms and downsize the text.
6. Conclusion section is missing from the manuscript.
7. It is strongly recommended to add a section on recommendations and future direction.
Reviewer 2 Report
Major concerns:
1) The introduction section is too long. Please abbreviate, for instance, thirst four paragraphs could be concentrated into one. 4, 5 and 6 paragraphs could be collided as well. The novelty of the present study remains unclear.
“The workflow is depicted in Figure 1. Firstly,..“ this paragraph could be depicted in result section.
2) Results start from figure 2. Please start from figure 1.
3) Figures are poorly prepared (and hard to read/understand). Figure legends are absent. Tables are difficult to read and could be depicted as supplementary files.
4) Discussion section is too short compared to introduction. There are no conclusions of the present results.
5) Does the prediction of some compounds suggested in the present study has any clinical value?
Taken together, the topic addressed in the present study is interesting and clinically important. However, the present manuscript must be improved significantly, and value of the present drug prediction model must be evaluated using some lab techniques, at least in vitro. Otherwise, the study adds no scientific significance or is impossible to verify.
Round 2
Reviewer 2 Report
Figures 2,3 and 6 must be improved. It is still unreadable.
Author Response
Figures 2,3 and 6 must be improved. It is still unreadable.
Response: Thanks for your kindly suggestion. We have improved the quality of these three figures with at least 300 dpi. Meanwhile, we added more description in the legends of these figures for helping understand.